# The Non-Coding RNome Landscape in Erythropoiesis: Pathophysiological Implications

**DOI:** 10.3390/cells14241971

**Published:** 2025-12-11

**Authors:** Emma Brisot, Laurent Metzinger, Valérie Metzinger-Le Meuth

**Affiliations:** 1HEMATIM UR-UPJV 4666, C.U.R.S, University of Picardie Jules Verne, CEDEX 1, 80025 Amiens, France; emma.brisot@u-picardie.fr (E.B.); laurent.metzinger@u-picardie.fr (L.M.); 2INSERM UMRS 1148, Laboratory for Vascular Translational Science (LVTS), UFR SMBH, University Sorbonne Paris Nord, 93000 Bobigny, France

**Keywords:** microRNA, long non-coding RNA, erythropoiesis, red blood cell, biomarker, gene regulation

## Abstract

**Highlights:**

**What are the main findings?**
Non-coding RNAs, including miRNAs and lncRNAs, play central regulatory roles in multiple stages of erythropoiesis, influencing lineage commitment, differentiation, maturation, hemoglobin switching, iron metabolism, and erythrocyte morphology.Specific miRNAs and lncRNAs modulate both transcriptional and post-transcriptional mechanisms, affecting processes such as chromatin remodeling, alternative splicing, apoptosis, enucleation, and erythroid-specific gene expression.Environmental and pathological signals reshape ncRNA expression patterns, further modulating erythropoiesis under stress and disease conditions.

**What are the implications of the main findings?**
Understanding ncRNA-mediated regulation of erythropoiesis opens new avenues for diagnostics, positioning distinct ncRNAs as potential biomarkers for anemia and other hematologic disorders.Therapeutic targeting of miRNAs and lncRNAs may enable precision modulation of erythropoiesis, offering novel treatment strategies for conditions involving ineffective red blood cell production.Mapping the non-coding RNome provides mechanistic insights that may improve interpretation of erythroid pathologies and inform the development of ncRNA-based clinical interventions.

**Abstract:**

Erythropoiesis is a multistage process critical for red blood cell production and systemic oxygen transport. It is tightly regulated, and recent advances have highlighted the pivotal regulatory roles of non-coding RNAs (ncRNAs), particularly microRNAs (miRNAs) and long non-coding RNAs (lncRNAs), in governing both physiological and pathological erythropoiesis. These ncRNAs have roles in the fine-tuning of the classical transcriptional and post-transcriptional control. This review explores the complex landscape of the non-coding RNome in erythroid differentiation, maturation, and function. We summarize how specific miRNAs influence erythroid lineage commitment, hemoglobin switching, iron metabolism, and cellular morphology, as well as their modulation by environmental and pathological cues. We also discuss emerging evidence on lncRNAs regulating chromatin remodeling, alternative splicing, apoptosis, enucleation, and erythroid-specific gene expression. These insights suggest that ncRNAs are instrumental orchestrators of erythropoiesis and accordingly, potential biomarkers and therapeutic targets in anemia and related hematologic disorders.

## 1. Normal and Pathological Erythropoiesis

Erythropoiesis is a complex, dynamic, and extremely well-regulated process in humans. It allows for the production of 2 × 10^6^ red blood cells (RBCs) per day under physiological conditions [1]. Erythrocytes are the most numerous cells in circulation [2] and are the only ones capable of transporting oxygen throughout the body. The production of erythrocytes must remain constant, thus allowing the maintenance of homeostasis [1]. In the event of significant blood loss or hypoxia, the kidneys release erythropoietin (EPO) to activate erythropoiesis and thus replenish the stock of RBCs [2,3]. This erythropoiesis begins in the bone marrow through the differentiation of hematopoietic stem cells (HSC) into Erythroid Bursting-forming units (BFU-E). Then these cells progress to the Erythroid Colony-forming units (CFU-E) stage. It is at this stage that the EPO receptor (EPOR) is active. Subsequently, in the presence of EPO, the cells transit from the CFU-E stage to the proerythroblast stage. The proerythroblast cell, in the presence of iron, becomes an erythroblast, then a reticulocyte, and finally a mature and functional RBCs (Figure 1) [1]. It is at the proerythroblast stage that heme synthesis occurs, thus allowing the formation of hemoglobin [4]. At the end of the process, after 120 days, RBCs become senescent, change morphology, and are naturally eliminated from circulation by macrophages [5]. The balance between the production and destruction of RBCs is therefore very tightly regulated [6]. In some cases, the erythropoiesis pathway may be inhibited by certain pro-inflammatory cytokines, for example, during inflammation or infection. In this case, stress erythropoiesis is activated. This operates differently from basal erythropoiesis and allows for a “burst of erythrocytes” to be produced, enabling the system to return to a state of homeostasis. However, this alternative system is active only temporarily. When its capacities are exceeded, for example, during chronic inflammatory diseases, the number of erythrocytes decreases, concomitant with enhancement of inflammatory cells, which can lead to the onset of anemia [6,7].

The number of erythrocytes is not the only element to take into consideration for the proper progression of erythropoiesis. Indeed, for an erythroblast to become mature and functional, it must also undergo a step of chromatin condensation, hemoglobinization, enucleation, and acquire a typical morphology, notably thanks to the presence of some specific cytoskeleton proteins such as erythrocyte tropomodulin of 41 kDa (E-Tmod41), which facilitates the organization of the cytoskeleton, for instance by organizing actin filaments (F-actin) [8,9,10].

### Regulation of Erythropoiesis

Erythropoiesis is an extremely well-regulated process. The slightest change or disturbance in the phenotype of RBCs can lead to various hematological disorders such as β-thalassemia, sideroblastic anemia [11], or anemias related to other pathologies such as chronic kidney disease (CKD) or several cancers. Since the phenotype of the cells is controlled by gene expression, a modification of their expression can occur at different levels. Among these regulators of gene expression, transcription factors such as GATA Binding Protein 2 (GATA2), GATA Binding Protein 1 (GATA1), and Krueppel-like factor 1 (KLF1), are the most studied [12]. However, new regulators are emerging and attracting increasing attention. Among these new regulators, non-coding RNAs (ncRNAs) are noteworthy, including microRNAs (miRNAs) and more recently long non-coding RNAs (lncRNAs). It has been shown that transcription factors and ncRNAs can act in a collaborative way at several levels to regulate gene expression during hematologic differentiation [12,13] (Figure 2). Please note that during erythroid differentiation, this epigenetic regulation is stage-specific. Indeed, some regulators will be highly expressed during early erythropoiesis, while others will be expressed during late erythropoiesis [14]. It has been shown that this ncRNA-dependent regulation occurs as early as the HSC stage. For example, depending on the ncRNA expressed, the cell will either become an erythrocyte, a macrophage, or a lymphocyte [15]. Under physiological conditions, all the different actors are perfectly synchronized and mechanistic studies (Table 1) show that the expression of the ncRNAs is causal in the orchestration of the proper progression of the process.

## 2. Non-Coding RNAs in Erythropoiesis

### 2.1. Non-Coding RNAs

It was in the 1990s, mostly with the development of high-throughput sequencing methods, that the majority of ncRNAs were discovered. This discovery led to the emergence of a new approach to genetics asserting, in the early 21st century, that more than 80% of the human genome codes for ncRNAs, whereas less than 3% of the genome codes for proteins [16].

Even though recent studies show that lncRNAs can occasionally encode small peptides, most ncRNAs do not code for proteins. Instead, they play a role in regulating gene expression [15]. We will focus on two main categories of ncRNAs: miRNAs, which are among the most studied ncRNAs currently, and lncRNAs, whose number has vastly expanded in recent years.

Currently, ncRNAs are being studied in all areas of health. For example, they can be found in oncology [17], neurology [18], cardiovascular [19], nephrology [20], hematology fields, and many others. Understanding their mechanism of action, as well as determining their potential targets (through algorithms such as TargetScan (https://www.targetscan.org/vert_80/ accessed on 5 December 2025) or Noncode (http://www.noncode.org/ accessed on 5 December 2025) [21], could allow these ncRNAs to be used as potential biomarkers, as well as therapeutic agents, particularly in the development of personalized medicine. In this review, we will focus more specifically on ncRNAs in the field of erythropoiesis. Several reviews have already addressed the role of miRNAs and/or lncRNAs in the differentiation and maturation of erythroid cells [10,22,23,24,25,26]. Our review aims to complement and update this literature by integrating recent discoveries and simultaneously addressing the roles of both miRNAs and lncRNAs in erythropoiesis, mostly from a mechanistic standpoint.

### 2.2. miRNAs and Erythropoiesis

The first miRNA, lin-4, was discovered in 1993 by Victor Ambros and Gary Ruvkun, but it was not until 2002 that a role for miRNAs in pathology was suggested, initially in cancer, and then in 2004 in hematopoietic diseases, among others [27]. Since then, miRNAs have been studied in more detail and their field of application has expanded to many areas [28]. Their importance in the gene regulation field explains the 2024 Nobel Prize in Physiology and Medicine [29]. It is well established that miRNAs are small non-coding RNAs of approximately 25 nucleotides, which regulate gene expression at the post-transcriptional level by binding to the 3′UTR part of the mRNA. This binding will lead to inhibition of translation or degradation of the mRNA. In this review, the details of the biogenesis of miRNAs will not be discussed (Figure 3) [28,30,31]. Thus, miRNAs can be involved in the regulation of genes controlling apoptosis, proliferation, differentiation, and cell metastasis [28,30]. For example, miR144/451 knockout (KO) mice present severe anemia with alteration of erythropoiesis, including apoptosis of erythroblasts [32]. In biological fluids, miRNAs can also be detected within microparticles and exosomes, allowing their use as biomarkers in serum, plasma, urine, and saliva [28,33]. For example, an increase in plasma levels of miR-451 could be associated with the degree of hemolysis in patients with β-thalassemia/hemoglobin E (β-thal/HbE), allowing it to serve as a potential biomarker for intravascular hemolysis in these patients [34].

It is well established that miRNAs play an essential role in regulating all stages of hematopoiesis [3]. Their differential expression enables them to have a functional role in both normal and malignant hematology [35]. Thanks to sequencing methods and databases such as miRBase (https://www.mirbase.org/ accessed on 5 December 2025) (miRBase: from microRNA sequences to function), numerous erythropoietic miRNAs have been identified. This has led to the emergence of studying miRNAs individually, giving clues to understanding their functioning and interaction with other elements of erythropoiesis.

#### 2.2.1. microRNA and Environmental Conditions

Environmental conditions or physiological status can influence the expression of miRNAs. The erythropoiesis process is tightly regulated and sensitive to external stimuli. Under stress conditions, such as hypoxia, oxidative stress, inflammation, some drugs or uremic toxins (UTs), the expression of specific miRNAs can be altered, leading to dysregulation of the entire process. For example, during erythroid differentiation under hypoxic conditions, overexpression of GATA1 and miR-210-3p is observed, which will in turn induce erythroid differentiation. The role of miR-210 is mediated by its target SMAD family member 2 (SMAD 2) [3,36] (Table 1). Similarly, miR-486 is sensitive to hypoxia and regulated by GATA1. It is involved in hypoxia-induced erythroid differentiation by targeting Silent information regulator sirtuin 1 (Sirt1) [37] (Table 1). In addition to hypoxia, other environmental conditions such as infectious stress can alter miRNA expression. For example, miR-22 regulates the erythroid/megakaryocyte balance, which can lead to infection-related anemia [38] (Table 1). Some medications, such as metformin, can also regulate miRNA expression. For example, in patients with Diamond Blackfan anemia (DBA), the expression of miR-26a is increased in response to metformin, leading to the blockade of Nemo-like kinase (NLK kinase) and promotion of erythroid differentiation [39] (Table 1). On the other hand, several miRNAs have their expression altered under oxidative stress conditions. This is the case for miR-214. Under physiological conditions, miR-214 expression increases during erythroid differentiation. However, in some pathologies such as β-thal/HbE or hemoglobin H (HbH), its expression is markedly altered. Indeed, in these hematological disorders, erythroid cells undergo oxidative stress, thereby disrupting normal erythropoiesis. It has been shown that in β-thal/HbE and HbH cells, a significant increase in miR-214 expression, caused by oxidative stress, leads to the repression of the antioxidant transcription factor Activation transcription factor 4 (ATF4). Since ATF4 is a target of miR-214, an increase in this miRNA results in the repression of this transcription factor, consequently reducing glutathione production and leading to the accumulation of reactive oxygen species (ROS) within the cell [40] (Table 1). UTs, which increase in the blood in CKD, can also disrupt normal erythropoiesis and contribute to the development of anemia. In the case of CKD, the kidneys are no longer functional and therefore cause these toxins to accumulate. UTs are very numerous and are known to have deleterious effects on many organs and tissues. These effects include oxidative stress, chronic inflammation, and disruption of cellular differentiation. These toxins are classified and listed by the European Uremic Toxins (EUTox). Among the 146 listed, the most studied include Indoxyl Sulfate (IS), p-cresyl sulfate and creatinine [41,42]. A number of miRNAs have their expression altered in the presence of IS, leading to dysregulation of erythropoiesis. One of these miRNAs is miR-223, which is known to play a regulatory role in inflammation, erythropoiesis, and cancers. The expression of miR-223 has been shown to decrease during normal erythropoiesis and increases in white blood cell differentiation [35]. Additionally, both miR-223 and its target LIM domain-only protein 2 (LMO2) are deregulated in the presence of IS in UT7/EPO cells and CD34+ progenitors [43]. This highlights its importance in the proper regulation of erythropoiesis. Moreover, factors such as different UTs, EPO levels, and dialysis can modify this miRNA expression [43] (Table 1).

#### 2.2.2. microRNAs and Globin Genes

Globin genes are essential for the formation of tetrameric hemoglobin. During the developmental process, the composition of hemoglobin changes. At the embryonic stage, the predominant form is embryonic hemoglobin (ζ_2_ε_2_). A first switch then allows the transition from embryonic hemoglobin to fetal hemoglobin, composed of α_2_γ_2_ subunits. In adults, a second switch leads to the formation of adult hemoglobin, composed of α_2_β_2_ subunits. These different switches are the result of a complex regulatory process involving both epigenetic and transcriptional regulators. When this process is dysregulated, several pathologies will develop, such as sickle cell anemia and β-thalassemia [44]. Among these regulators, miRNAs are preponderant. Increasing evidence shows that these RNAs are essential for the regulation of globin genes. By regulating levels of globin genes like γ and β, miRNAs ensure that the globin switch occurs when necessary. When the expression of these miRNAs is dysregulated, hemopathies arise. This is notably the case for miR-210, which indirectly increases the expression of the γ-globin gene during erythroid differentiation. Indeed, miR-210 is targeted by phospholipase C β1 (PLCβ1). KO of PLCβ1 expression induces an increase in miR-210 levels and an upregulation of γ-globin expression via the protein kinase C α (PKCα) signaling pathway [45] (Table 1). Other miRNAs are also important for regulating γ-globin, such as miR-34a, which increases Hemoglobin F (HbF) levels. The persistence of HbF in children and adults points to sickle cell anemia. It has been shown that overexpression of miR-34a leads to reduced expression of signal transducer and activator of transcription 3 (STAT3) phosphorylation, in turn inhibiting γ-globin gene expression [46]. Conversely, blocking STAT3 gene expression indirectly activates γ-globin through miR-34a [47] (Table 1).

During erythroid differentiation, miR-23a and -27a are involved in the regulation of the β-globin gene. Specifically, during erythroid differentiation, miR-23a and miR-27a promote β-globin gene transcription by repressing Krüppel like factors 3 (KLF3) and Specificity Protein 1 (SP1) transcription factors. On the other hand, KLF3 can bind to the miR-23a/miR-27a cluster, inhibiting the expression of these miRNAs and consequently suppressing β-globin gene expression. These results highlight the presence of a positive feedback loop regulation system [48] (Table 1).

During the γ- to β-globin switch, miR-326 plays an important role by interacting with the 3′UTR end of the KLF1 transcription factor mRNA. Expression of miR-326 decreases KLF1 expression, leading to increased γ-globin gene expression. Additionally, Li and his team have shown that the relative expression of miR-326 is positively correlated with HbF levels in patients with β-thalassemia [49] (Table 1). This is also the case for miRNA-2355-5p, which correlates with γ-globin synthesis by suppressing Kruppel-like factor 6 (KLF6) transcription factor expression [50] (Table 1).

**Table 1 cells-14-01971-t001:** microRNA (miRNAs) and erythropoiesis. Selected miRNAs involved in erythropoiesis are listed in this table.

microRNA	Species	Function	Mechanisms/Regulation	Related Pathology	References
miR-210-3p	human	increase of erythroid differentiation	SMAD2	Hypoxic conditions	[3,36]
miR-486	human	Hypoxia-induced erythroid differentiation	Sirt1	Hypoxic conditions	[37]
miR-22	human	Erythroid/megakaryocyte balance	Tet1	Inflammatory conditions and myeloid malignancies	[38]
miR-26a	murine	Promoting erythroid differentiation	NLK	Diamond Blackfan anemia and Metformin	[39]
miR-214	human	Glutathione production and accumulation of ROS, disrupting normal erythropoiesis	ATF4	Oxidative stress and β-thalassemia	[40]
miR-223	human	Dysregulation of erythropoiesis	LMO2	Uremic Toxins and CKD	[43]
miR-210	human	Increases the expression of γ-globin gene during erythroid differentiation	PLCβ1	β-thalassemia	[45]
miR-34a	human	Inhibiting γ-globin gene	STAT3	Sickle cell anemia	[46,47]
miR-23a and miR-27a	human	Regulation of β-globin gene	KLF3 and SP1	ND	[48]
miR-326	human	Switch γ to β-globin	KLF1	β-thalassemia	[49]
miR-2355-5p	human	γ-globin synthesis	KLF6	β-thalassemia and Sickle cell disease	[50]
miR-218	human	Iron metabolism	ALAS2	Iron toxicity	[51]

From one species to another, miRNAs can vary, and their dysregulation can lead to the deregulation of several actors. This change in expression can then lead to the onset of diseases. Conversely, some diseases are also caused by the dysregulation of miRNA expression. SMAD2: SMAD family member 2; Sirt1: Silent information regulator sirtuin 1; NLK: Nemo-like kinase; ATF4: Activation transcription factor 4; LMO2: LIM domain-only protein 2; PLCβ1: phospholipase C β1; STAT3: signal transducer and activator of transcription 3; KLF3: Krüppel-like factors 3; SP1: Specificity Protein 1; KLF1: Kruppel-like factor 1; KLF6: Kruppel-like factor 6; ALAS2: 5′-aminolevulinate synthase 2; ND: Not determined. All studies were mechanistic.

#### 2.2.3. microRNA and Iron Homeostasis

Iron is an essential element for the proper process of erythropoiesis. Erythroblasts require large amounts of iron to enable heme synthesis, and thus use approximately 80% of plasma iron [52]. Since iron can be toxic to the body when present in excess, its regulation is instrumental and involves several key actors. Iron uptake, storage and utilization are regulated in part by miRNAs, such as miR-320, miR-485-3p, miR-584 or miR-218, which regulate the expression of several genes associated with iron homeostasis mechanisms, thereby contributing to the maintenance of physiological iron concentrations [53]. For instance, miR-218 is involved in erythroid differentiation and regulation of iron metabolism, by targeting the 3′UTR end of the 5′-aminolevulinate synthase 2 (*ALAS2*) gene, encoding the first enzyme involved in heme synthesis. Overexpression of miR-218 thus inhibits erythroid differentiation and alters iron metabolism, leading to iron accumulation in mitochondria [51] (Table 1).

#### 2.2.4. miRNAs and Regulation of the Different Stages of Erythropoiesis

During erythroid differentiation, miRNA involvement will impact either early or late erythropoiesis. Dysregulation of miRNA expression during early erythropoiesis is more likely to cause a phenotypic change in the cell, while deregulation during late stages will lead to cell death.

Terminal erythroid differentiation is regulated by miR-451 through its influence on the expression of many relevant genes. Its expression increases during erythroid differentiation through an Ago2-dependent maturation pathway [9] (Figure 4). Terminal erythroid differentiation is also repressed by miR-150, whose expression decreases during the final stages of erythropoiesis. Overexpression of miR-150 inhibits erythroid differentiation and leads to apoptosis, cell cycle arrest, and reduced expression of hemoglobin genes by suppressing the ErbB-MAPK-p38 and ErbB-PI3K-AKT pathways, through targeting protein 4.1R (EPB41, a RBC membrane protein) [54]. Interestingly, miR-150 is indirectly regulated by the ferritin heavy subunit (FHC), which affects erythropoiesis and miR-150 function [55] (Figure 4).

The decrease in miR-191 expression is essential for chromatin condensation and enucleation. In the murine model of erythropoiesis, miR-191 expression decreases from the CFU-E stage onward. This reduction allows the activation of its targets Riok3 and Mxi1, leading to the inhibition of the histone acetyltransferase Gcn5, thereby facilitating chromatin condensation and enucleation [56] (Figure 4). 

The process of terminal erythropoiesis includes contributions from miR-669m. Its transgenic overexpression has been shown by Kotaki et al. to decrease erythroid differentiation markedly in a murine model during late stages of erythropoiesis. The specific targets of miR-669m are A-kinase anchoring protein 7 (Akap7) and X-linked Kx blood group (Xk) in the RBC. Suppression of Akap7 expression leads to impaired heme biosynthesis, while alteration of Xk gene primarily modifies the RBC membrane complex. Interestingly, miR-669m is not constitutively expressed in RBC; however, in cases of dysregulation of its expression, alterations in late erythropoiesis have been described [57].

Some miRNAs are involved in both early and late erythropoiesis. This is the case for miR-23a, which is involved in early erythropoiesis due to its role in erythroid differentiation. It has been shown by Zhu et al. that this miRNA promotes erythroid differentiation in the erythropoietic K562 cell model as well as in CD34+ primary hematopoietic cells. However, when erythroid cells differentiate, an increase in GATA1 expression and miR-23a accumulation is observed. Subsequently, miR-23a represses Src homology-2 domain-containing protein tyrosine phosphatase 2 (SHP2), a tyrosine phosphatase acting as a negative regulator of erythroid differentiation, to promote terminal erythropoiesis. Inhibition of miR-23a prevents the maturation of erythroblasts. Moreover, ectopic expression of miR-23a leads to the accumulation of mature erythroid cells and the formation of “erythroid clones” in primary CD34+ hematopoietic progenitor cell cultures. Thus, the participation of miR-23a to the GATA-1 and SHP2 complex makes this ncRNA an important regulator of hematopoietic lineage specification [21] (Figure 4).

Other miRNAs are expressed in undifferentiated progenitors during early erythropoiesis. This is notably the case for miR-155, which is predominantly expressed at undifferentiated stages, while its expression decreases when cells begin to differentiate. Downregulation of miR-155 is thus likely necessary for erythroid differentiation [58].

It has been shown that the miR-221 and -222 genes are closely located and share a common promoter [30]. They are highly expressed in undifferentiated progenitors, and their expression decreases during differentiation and erythroid maturation. When their expression levels decrease during erythropoiesis, functional proteins, including Kit protein, are activated. Indeed, miR-221 and miR-222 can directly interact with the 3′UTR end of Kit mRNA [59]. Kit protein is, however, not the only target of these miRNAs. For example, Scaffold/Matrix attachment region binding protein (SMAR1), a protein involved in erythroid differentiation, can also regulate the expression of miR-221 and miR-222 [30]. Additionally, it is interesting to note that SMAR1 can also be targeted by miR-320a. When miR-320a expression decreases, an increase in SMAR1 protein expression is observed, along with a reduction in its targets, Bcl-2-associated X protein (BAX) and p53 up-regulated modulator of apoptosis (Puma), leading to inhibition of apoptosis [30]. A study by Jiang et al. shows that miR-222 also targets Biliverdin Reductase A (BLVRA) and CT10 regulator of kinase like (CRK-like), two proteins involved in erythroid differentiation. Thus, downregulation of BLVRA and CRKL by miR-222 is involved in erythroid differentiation [60] (Figure 4). These studies emphasize that one target gene can be regulated by different miRNAs and vice versa in RBCs as well as other cell types.

During early stages of erythropoiesis, miR-223 is highly expressed. In later stages, its expression decreases, allowing a higher expression of its target protein LMO2 (Figure 4). In contrast, miR-223 expression increases during megakaryocytic and granulocytic differentiation [35]. As previously mentioned, this miRNA plays a crucial role in the proper progression of erythropoiesis. It can be easily deregulated by disruptive elements, such as UTs. Consequently, any disturbance of this miRNA can lead to dysregulation of the erythropoiesis process [43]. In addition, Vian et al. showed that variation in miR-223 expression can directly impact the fate of hematopoietic stem cells during differentiation [61]. Depending on the transcription factor involved and regulatory region of its pri-miR precursor, miR-223 expression will be modified, and consequently, the cell fate will differ. For example, when the transcription factor CCAAT/enhancer binding protein α (C/EBPα) binds to the distal region of pri-miR-223, the cell will favor a monocyte profile. However, if this same transcription factor binds to the proximal region of pri-miR-223, the cell will tend towards a granulocyte phenotype. Finally, when the transcription factors T-cell acute lymphocytic leukemia protein 1 (TAL1), LMO2, and GATA1 bind the proximal region, this represses the expression of miR-223, thereby allowing erythrocyte differentiation. Thus, it is demonstrated that recruitment and action of different transcription factors, specific to each lineage, on two regulatory regions of miR-223 precisely modulate the transcription of two different precursors. This study exemplifies that different regulatory regions of miRNA expression can be used as an additional means to participate in cellular diversity [61].

The inhibition of erythropoiesis involves miR-9 as a key regulator. Its ectopic expression induces ROS production within erythroid progenitors, thus inhibiting their differentiation. For this, miR-9 induces the downregulation of the transcription factor fork-head box O3 (FoxO3) by direct binding to its 3′-UTR region, also leading to the downregulation of its target genes B cell translocation gene 1 (Btg1) and CBP-p300 interacting transactivator with Glu/Asp-rich carboxy-terminal domain 2 (Cited2) (Figure 4). Accordingly, Zhang and his team demonstrated that the expression of FoxO3-3A, an activated form of FoxO3, can reverse the inhibition of erythroid differentiation induced by miR-9 [62].

Mice lacking miR-144/451 suffer from blocked erythroid differentiation, with increased c-Myc expression. The miR-144/451 cluster is involved in erythroid differentiation by targeting the transcription factor c-Myc. During erythroid differentiation, GATA1 expression increases, while, conversely, Myc protein levels decrease. This increase in GATA1 enables the activation of miR-144/451 expression, thereby blocking c-Myc expression at the post-transcriptional level. This regulatory process promotes erythroid differentiation. Additionally, GATA1 directly inhibits c-Myc and a positive regulator of miR-144/451 (Figure 4). Thus, there is a link between GATA1-miR-144/451-Myc in the regulation of erythroid differentiation [63]. Another miRNA, miR-152, also represses GATA1 expression during erythropoiesis [64] (Figure 4).

The transcription factor c-Myc is also a target of miR-155 (Figure 4). In 2024, Penglong et al. suggested that this miRNA regulates the ineffective erythropoiesis of β-thalassemia by modulating the expression of the c-myc gene. This altered expression allows for the control of erythroblast proliferation and differentiation. Indeed, it was shown that in β-thalassemia, the overexpression of miR-155 increases the number of basophilic and polychromatic erythroblasts [65].

Erythropoiesis is also supported by miR-199b-5p, which promotes the differentiation of erythroblast precursors into erythroblasts. This miRNA is regulated by the transcription factors GATA1 and Nuclear Factor-Erythroid 2 (NF-E2), enabling the activation of its transcription. Expression of miR-199b-5p represses the expression of its target protein c-kit, thus promoting erythroblast maturation. There are also other potential target genes of miR-199b-5p, such as TATA-box binding protein associated factor 9B (TAF9B) and Cyclin L1 (CCNL1) (Figure 4) [66].

The above-mentioned miRNAs can function in a coordinate manner during erythropoiesis. For example, the underexpression of miR-155 precedes the overexpression of miR-451 in undifferentiated cells on day 6 of erythroid differentiation (Figure 4) [58]. Similarly, miR-320a negatively regulates the SMAR1 protein, and its expression is associated with increased levels of miR-221 and miR-222 (Figure 4) [30].

In addition to their individual roles, some miRNAs can function in complexes, such as the miR-99a/100 and miR-125b, which regulate the transforming growth-factor-β (TGFβ) and Wnt pathways (Figure 4). These miRNAs are produced from a polycistronic pri-miRNA transactivated by the stem cell regulator Homeobox A10 (HOXA10) [67]. Similarly, miR-99b/let-7/125a tricistron, located on chromosome 19, plays a role in regulating erythroid differentiation and is involved in several pathologies such as myelodysplastic syndrome [68].

Finally, miR-126 and miR-150 regulate the c-Myb protein, which is essential for the hematopoietic process (Figure 4). Studies on animal models show that alterations in these miRNAs can lead to abnormalities in erythrocyte and thrombocyte formation [69]. These results highlight the importance of a precise miRNA regulation network in cellular development and differentiation.

The biogenesis of miRNAs [31] is an important step for the proper progression of erythropoiesis. During erythroid differentiation, the expression of the original precursor pri-124-1 increases while the expression of later precursor pre-124-1 and mature miR-124 decreases. Erythropoiesis is negatively regulated by miR-124, which inhibits erythroid differentiation through its targets c-Myb and TAL1. This miRNA functions with the RNA-binding protein QKI5 (a protein belonging to the RNA-binding protein family that decreases during erythroid differentiation) (Figure 4). This long-distance RNA-RNA interaction works through QKI5’s recognition of a QRE (QKI response element) motif in the cis-distal region of pri-124-1. QKI5 also recruits the microprocessor to recognize and cleave the pri-miRNA to obtain the pre-miRNA precursor. Overexpression of QKI5 has been shown to reduce pri-124-1 but to increase miR-124 [70].

Some miRNAs have their expression altered in erythropoiesis-related pathologies, such as miR-101-3p. Indeed, its overexpression was detected in erythroblasts of patients with β-thal/HbE. Phannasil et al., based on a correlation study, highlighted that the expression level of this miRNA varies depending on the stages of erythropoiesis, since an increase in the expression of miR-101 was observed in erythroblasts from β-thal/HbE patients [71]. In DBA, a blockade of erythropoiesis occurs between the BFU-E and CFU-E stages, and an RPS19 mutation is found in 25% of patients. At the same time, the Special AT-rich sequence-binding protein 1 (SATB1) is downregulated when miR-30 and -34 are upregulated. In DBA pathology with RPS19 deficiency, SATB1 is downregulated due to upregulation of miR-30 and miR-34, which bind the 3′UTR of SATB1 mRNA. Overexpression of miR-34 occurs when p53 is upregulated. This shows that increased levels of p53 lead to the overexpression of miR-34, which then cooperates with miR-30 to downregulate SATB1. However, unlike miR-34, miR-30 is not directly regulated by p53 [72].

#### 2.2.5. miRNAs and Erythrocyte Morphology

To fulfill their functions, erythrocytes display a specific morphology acquired in the course of erythropoiesis. This process is tightly regulated by a number of miRNAs that ensure the acquisition of the erythroid biconcave shape, the integrity of the erythrocyte membrane and proper nuclear expulsion (Figure 5). This is the case for miR-142, miR-23b and miR-144. By regulating F-actin, miR-142 plays a crucial role in maintaining the biconcave shape of erythrocytes and organizing the membrane cytoskeleton (Figure 5) [2]. The cytoskeleton is indeed instrumental in maintaining the characteristic biconcave shape of the RBC. This regulation contributes to keeping the size of RBCs, thereby ensuring the physico-mechanical properties of erythroblasts and guaranteeing the lifespan and number of erythrocytes. The morphology of RBCs is influenced by miR-23b, which regulates the expression of genes encoding membrane cytoskeleton proteins during erythroid differentiation induced by “fluid shear stress.” It targets the actin-binding E-Tmod41 protein (a protein binding to F-actin) at its 3′UTR end, and its expression is inhibited during Shear Stress, resulting in increased expression of E-Tmod41 (Figure 5) [8].

To obtain a mature and functional erythrocyte, a chromatin condensation step is necessary to expel the cell nucleus. This maturation step is ensured by the key role of miR-144. During erythroid differentiation, the chromatin-associated high mobility group protein N2 (HMGN2) maintains the chromatin in a decondensed state, preventing erythroblast maturation. In the early stages of this differentiation, miR-144 binds to the 3′-UTR region of HMGN2 mRNA, thereby blocking its expression and allowing chromatin condensation (Figure 5) [73].

### 2.3. lncRNAs and Erythropoiesis

Although lncRNAs have been less studied than miRNAs, their discovery dates back to the 20th century. The first lncRNA studied was the X-inactive specific transcript (XIST) in 1990 [74]. It is known to play a major role in X-chromosome inactivation in females. Other lncRNAs, such as HOX transcript antisense RNA (HOTAIR), metastasis-associated lung adenocarcinoma transcript-1 (MALAT1), urothelial cancer associated 1 (UCA1), nuclear paraspeckle assembly transcript 1 (NEAT1), etc., were discovered between 1990 and 2003, thanks to the Human Genome Project [75]. However, thanks to improved database analyses, a study published in 2024 identified 140,268 new human lncRNA transcripts in the GENCODE database [76]. Due to their recent discovery, the number of studies on these ncRNAs is scarcer when compared to miRNA-related studies [77]. A few recent dedicated databases list a significant number of lncRNAs, such as LNCipedia (lncipedia.org) or LNCBOOK (ngdc.cncb.ac.cn/lncbook/home) [78]. Thus, these lncRNAs are increasingly studied, with about 50,000 publications on “lncRNAs” and over 2000 publications reporting their function [79]. However, clarifying the functions of lncRNAs seems more complex than detecting them [80]. Like miRNAs, lncRNAs are involved in neurology, oncology, cardiovascular pathologies [75], hematology, and so on [81].

It is well established that lncRNAs are defined as transcripts of more than 200 nts, transcribed by polymerase II, with a 5′ end, polyadenylated tail, and splicing [80]. Their expression levels are lower than those of protein-coding genes and are more specific to a particular cell type [15]. However, due to their structural diversity, the mechanisms of action of lncRNAs are more varied and complex than those of miRNAs [15]. They can act in the nucleus, where they are involved in chromatin remodeling, transcription, pre-mRNA splicing, and RNA stability (Figure 6) [15]. In the cytoplasm, they can bind to anchoring proteins or to RNA as “microRNA sponges,” thus inactivating them by binding them in a complementary manner and diverting them from their target mRNAs (Figure 6) [15]. Based on their genomic location relative to protein-coding genes, lncRNAs can be categorized into different subgroups such as intergenic (lincRNAs), transcribed from the same promoter as a protein-coding gene (pancRNA), associated with an enhancer (eRNA), and circular (circRNAs). Other categories of lncRNAs also exist, such as promoter-associated long non-coding RNAs (plnRNAs), antisense long non-coding RNAs (alncRNAs), short half-life long non-coding RNAs (shlncRNAs) and enhancer long non-coding RNAs (elncRNAs). However, some lncRNAs do not fit into these categories, demonstrating the complexity of studying these non-coding RNAs [10,82].

The diverse regulatory interactions of ncRNAs are increasingly recognized. While these interactions remain poorly characterized in hematopoiesis, evidence from other fields demonstrates that miRNAs and lncRNAs can mutually regulate each other through multiple mechanisms. Many lncRNAs act as competitive endogenous RNAs, binding miRNAs to prevent them from repressing target mRNAs. This sponging mechanism allows lncRNAs to indirectly increase the expression of specific protein-coding genes. The miRNAs can also bind to lncRNAs and promote their degradation, reducing the lncRNA’s regulatory activity [83]. Through mutual regulation, miRNAs and lncRNAs form feedback loops that fine-tune cellular signaling pathways. Deregulation of miRNA–lncRNA networks is linked to diseases such as cancer, cardiovascular disorders, and neurological conditions. These networks act as molecular buffers, maintaining gene-expression stability under stress [84].

The presence of lncRNAs in erythropoiesis was first demonstrated in 2011 by the Hu team. They discovered an anti-apoptotic role of the lincRNA erythroid pro-survival (LincRNA-EPS) in erythroid differentiation in a murine model [85]. Since then, through transcriptome studies on erythroblasts, megakaryocytes, and erythro-megakaryocytic precursors, 1109 lncRNAs expressed in mice (including 683 in erythroblasts) [81] and 594 in humans have been identified [10]. This study highlights that, in addition to being expressed during erythropoiesis, lncRNAs are species-specific, with only 15% common to both mice and humans [10]. However, conservation exists among different mouse strains [81]. As a result, some articles define lncRNAs as a “new class of erythropoiesis modulators” [81,86]. Like miRNAs, the roles and functions of lncRNAs are diverse, and their regulation is essential for the proper progression of erythropoiesis. When these RNAs are dysregulated, the entire balance is disrupted, potentially leading to the development of erythropoietic disorders. It should be mentioned that a limitation for the study of lncRNAs is their uniqueness to the human species, which presents a challenge in transposing murine findings to humans due to their lack of conservation. 

#### 2.3.1. lncRNAs and Environmental Conditions

Like most miRNAs, lncRNAs can have their expression altered by external conditions. This is the case for the lncRNA Hypoxia-Induced Kinase-mediated Erythropoietic Regulator, or LINC02228 (HIKER). Under hypoxic conditions, particularly in patients with Monge’s disease (mountain sickness, resulting in excessive erythropoiesis), the lncRNA HIKER and its target CSNK2B are overexpressed and upregulated. Azad et al. demonstrated that downregulation of HIKER decreases its target CSNK2B levels, leading to the reduction in excessive erythropoiesis (Table 2) [13].

#### 2.3.2. lncRNAs and Chromatin Regulation

Chromatin, along with its remodeling, plays a key role in the regulation of gene expression. Depending on its conformation and accessibility, a gene can be expressed at different levels. Chromatin remodeling represents one of the major epigenetic mechanisms involved in the control of gene expression. This process is tightly regulated by various epigenetic factors, including ncRNAs [87]. Erythroid differentiation can be regulated by a close association between lncRNAs, transcription factors such as GATA1, and chromatin remodeling. This is the case with the lncRNA PC-esterase domain-containing 1B antisense RNA 1 (PCED1B-AS1) (Table 2). Chromatin accessibility in erythroblasts allows GATA1 (through its binding to the distal region) to regulate the transcription of the lncRNA PCED1B-AS1, which in turn affects erythroid differentiation. Zhu et al. showed that the expression of this lncRNA gradually increases as erythroid differentiation progresses, reaching its maximum expression during late stages of differentiation. Although still poorly understood, this could reflect the importance of this lncRNA in the enucleation process [88]. Another study conducted by Ren et al. highlights the importance of chromatin accessibility, transcription factors, and lncRNAs during erythroid differentiation. The authors show in this transcriptomic study that these epigenetic changes are stage-specific during erythroid differentiation [14]. Two lncRNAs were highlighted, PCED1B-AS1 (described previously) and the lncRNA Anti-Differentiated Non-Coding RNA (DANCR). The DANCR lncRNA regulates erythropoiesis by controlling chromatin. The DANCR lncRNA is known to drive erythroid differentiation by altering megakaryocytic differentiation. This lncRNA functions in coordination with chromatin accessibility and transcription factors such as Runt-related transcription factor 1 (RUNX1) (Table 2) [14].

Since chromatin is important for the regulation of many genes, it also plays a key role in the regulation and activation of β-globin expression and thus performing the switch from HbF to adult hemoglobin. This complex regulation also involves the presence of lncRNAs. This is the case with the BGLT3 lncRNA. During the transition from HbF to HbA, the BCL11A factor binds to the γ-globin gene (HBG2/1 gene) to repress γ-globin expression. For complete repression of HBG2/1, the lymphoma/leukemia-related factor/ZBTB7A (LRF/ZBTB7A) transcription factor is necessary. Chondrou et al. observed that the binding of the LRF transcription factor to the BGLT3 gene activates BGLT3 expression. This lncRNA is produced in the γ-δ intergenic region of the β-globin gene locus, initiating the transcriptional events necessary for the switch from γ-globin to β-globin (Table 2). To express the β-globin gene, it was established that the chromatin conformation in this region was in an open form, allowing for the expression of the β-globin gene and thus the hemoglobin switch. This study highlighted that β-globin expression depends on BGLT3 [89].

Another example of a lncRNA involved in chromatin regulation is the lncRNA Bloodlinc (or alncRNA-EC7). This murine lncRNA acts as a super-enhancer for Solute carrier family 4, member 1 (SLC4A1) [90], the gene encoding Band3, the primary anion exchanger of erythropoiesis. The reach of this RNA is wider than its target site, notably through its transaction. This lncRNA impacts important regulators and effectors of terminal erythropoiesis and can directly bind to the chromatin attachment factor Heterogeneous nuclear ribonucleoprotein U (HNRNPU) (Table 2). Inhibition of Bloodlinc or HNRNPU alters the genetic program of erythropoiesis, blocking erythropoietic cell production. However, ectopic expression of Bloodlinc leads to the proliferation of erythroid progenitors, terminal erythropoiesis, and enucleation of reticulocytes even in the absence of differentiation stimuli. However, this lncRNA is not conserved between murine and human models, making its study particularly complex [10,81,91].

#### 2.3.3. lncRNAs and Regulation of Alternative Splicing

Alternative splicing is crucial for the regulation of gene expression in humans. Indeed, 95% of human genes undergo alternative splicing, contributing to the diversity of RNA transcripts and the proteins produced [92]. Alternative splicing not only regulates lncRNAs, but can also be actively regulated by them [92]. The alternative splicing of a target pre-mRNA can be modulated by lncRNAs through direct interaction with a splicing factor. This is the case with the lncRNA Fas antisense 1 (Saf or Fas-AS1). Villamizar et al. demonstrated that this lncRNA, located within the cell nucleus, interacts with the Human splicing factor 45 (SPF45), promoting the exclusion of exon 6 in the pre-messenger RNA of Fas. The exclusion of this exon subsequently allows the production of Fas in its soluble form. A KO of SPF45 leads to a decrease in the alternative splicing of Fas pre-messenger RNA, a reduction in its soluble form, an accumulation of Fas at the membrane level, and hence an increase in apoptosis. It is therefore the soluble form of Fas that blocks cellular apoptosis. This study conducted on erythrocytic cells shows that this regulatory mechanism could be generalized to different types of cells [93].

#### 2.3.4. lncRNAs and Apoptosis Regulation

Like other cellular processes, apoptosis (or programmed cell death) is tightly regulated. Some lncRNAs can, through their expression, block apoptosis and thus prevent cell death. The lncRNA Saf protects erythroblasts from apoptosis. During erythropoiesis, Saf is positively regulated by the transcription factors GATA1 and KLF1 and negatively regulated by nuclear factor-kappa B (NF-κB) (Table 2). NF-κB activity suppresses Saf transcription. When NF-κB expression decreases during erythropoiesis, lncRNA Saf takes over, preventing cells from entering apoptosis even in the presence of decreased levels of NF-κB. Saf interacts with Fas pre-mRNA by complementary base pairing. Saf also participates in the splicing of Fas pre-mRNA by excluding exon 6, facilitating the production of soluble Fas protein in order to protect cells from Fas-mediated apoptosis. This lncRNA allows the regulation of the Fas apoptotic pathway and thus reduces apoptosis of erythrocytic cells in humans once NF-κB is inactivated [77,93].

Located in the nucleus, lincRNA-EPS is involved in mouse erythropoiesis, notably by suppressing apoptosis via abolishing the expression of the ASC/Pycard protein (a signaling molecule that induces cell death by activating caspases) (Table 2). However, the precise mechanisms explaining how this lncRNA functions within erythropoietic cells are still poorly understood. Furthermore, although its usefulness could represent a major advancement in the treatment of anemia, its non-conservation in humans emphasizes the need to discover a yet undetected human homolog [85,94].

#### 2.3.5. lncRNAs and Heme Regulation

Heme regulation is very important as it is an essential component for hemoglobin formation. Like miRNAs, various lncRNAs are involved in the regulation of this process. Among the various regulatory agents, lncRNA UCA1 acts at the post-transcriptional level to regulate the stability of ALAS2 mRNA, the first enzyme involved in heme synthesis. This regulation is possible through the mobilization of an RNA-binding protein, polypyrimidine tract-binding protein 1 (PTBP1) (Table 2). The expression of UCA1 is maximal during the early stages of erythroid differentiation. Thus, a deficiency in the lncRNA UCA1 leads to inhibition of erythroid differentiation, particularly at the proerythroblast stage. Moreover, this inhibition can induce the dysregulation of ALAS2 mRNA and thus an alteration in heme biosynthesis [95].

#### 2.3.6. lncRNAs and Enucleation Regulation

Enucleation is very important for the proper progression of erythropoiesis as well as for the fate of erythroblasts. Indeed, although its process is still poorly understood, it is extremely well regulated. Thus, a number of lncRNAs, such as shlnc-EC6, are involved in the regulation of this process. The murine lncRNA shlnc-EC6 participates in the regulation of this process. Wang et al. showed an upregulation of this lncRNA in reticulocytes in the course of erythropoiesis. This study shows that a KO of shlnc-EC6 blocks erythroid enucleation by negatively regulating the expression of Ras-related C3 botulinum toxin substrate 1 (Rac1), an important RhoGTPase for cytoskeleton, survival, proliferation, adhesion, and cell migration, and phosphatidylinositol 4-phosphate 5-kinase (PIP5K) (Table 2). This negative regulation occurs at the post-transcriptional level, notably through the presence of specific binding sites in the 3′UTR of Rac1. The kinase PIP5K serves as an effector downstream of Rac1. This study thus highlights the role of shlnc-EC6 as a novel modulator of murine erythropoiesis through the Rac1/PIP5K signaling pathway. However, to date, no equivalent of this lncRNA has been found in humans [86].

#### 2.3.7. lncRNAs and Erythroid Differentiation

Like miRNAs, lncRNAs are important for erythroid differentiation, even though their involvement at the different stages is poorly understood. Furthermore, given their more recent discovery, it would not be surprising if, in the coming years, advances in new technologies lead to the identification of additional lncRNAs involved in the regulation of this process. Indeed, ongoing technological progress is already enabling the discovery of novel lncRNAs, suggesting that some of them may also play a role in erythropoiesis [71]. One example among these lncRNAs is NEAT1. This lncRNA is exclusively localized in paraspeckles, nuclear structures enabling genetic regulation, where it contributes to maintaining the integrity of the nuclear bodies [96]. Another specificity of this lncRNA is that it exists in two isoforms: NEAT1_1 and NEAT1_2. These two isoforms are transcribed from the same locus located on chromosome 11 in humans and 19 in mice. Differences in expression between these two isoforms have been described. For example, in murine hematopoietic stem cells, the NEAT1_1 isoform is preponderant compared to NEAT1_2 [97]. Although this lncRNA is conserved between mice and humans, its function in erythropoiesis seems disparate. Indeed, NEAT1 KO mice, under basal conditions, exhibit damage in spleen erythropoiesis, whereas bone marrow erythropoiesis is not affected. Furthermore, it is important to specify that deletion of the NEAT1 gene during erythropoiesis stress induced in the murine model does not affect this cellular process. However, in human CD34 cells derived from cord blood, it has been shown that reduction in NEAT1 lncRNA leads to impairment of erythropoiesis. NEAT1 is thus involved in erythropoiesis, known to bind different erythrocytic transcription factors such as GATA1, KLF1, and Myb in various tissues, but its exact role in erythropoiesis has to be further investigated (Table 2) [96].

Recently identified by Matur et al., in 2025 [98], lncRNA ncRNA-a3 is also essential for proper erythropoiesis. The authors demonstrate that this lncRNA acts as an eRNA, enabling the activation of the transcription factor TAL1 (Table 2). TAL1, in close cooperation with GATA1, LMO2 and LIM domain binding A (LDB1), modulates chromatin accessibility through E1 binding protein p300 and Brahma-related gene 1 (BRG1), allowing the activation of key genes involved in the self-renewal of hematopoietic stem cells and the commitment to hematopoietic lineages. A KO of ncRNA-a3 in erythropoietic K562 cells leads to a significant decrease in TAL1 levels, which is in discordance with results in T-leukemic Jurkat cells, suggesting an erythroid-specific function of this ncRNA. The KO of this lncRNA results in the accumulation of BFU-E, a reduction in CFU-E and glycophorin A (GPA)+ cells, confirming the importance of ncRNA-a3 in the commitment to erythroid differentiation. Furthermore, the study also showed that by altering chromatin accessibility, ncRNA-a3 affects the TAL1-mediated transcriptional program in erythroid cells, influencing the expression of GATA1, EPOR, HBG, GYP, TAL1 and LYL1. Thus, ncRNA-a3 is crucial for the differentiation and maturation of erythroblasts, hemoglobin regulation and heme biosynthesis [98].

Cpoxe RNA, involved in terminal erythropoiesis, is an eRNA derived from a super-enhancer (ncRNA transcripts originating from bidirectional transcriptional activity at enhancer regions). eRNAs are generally short-lived and contribute to the regulation of numerous genes [99]. CpoxeRNA is a nuclear lncRNA transcribed in the antisense orientation upstream of the Cpox gene, which encodes an enzyme involved in heme biosynthesis. KO of CpoxeRNA leads to impaired proliferation and enucleation of erythroblasts during terminal erythropoiesis. In contrast, KO of Cpox mRNA does not impact this process, suggesting that CpoxeRNA regulates erythropoiesis through other target genes or mechanisms independent of Cpox. CpoxeRNA functions as an active eRNA by modulating chromatin organization, notably through its interaction with the CTCF/cohesin complex (Table 2). This interaction promotes the formation of long-range chromatin loops, enabling the regulation of genes involved in erythropoiesis, particularly through trans-acting and indirect mechanisms. Note however that CpoxeRNA has only been characterized in rodents [100].

Finally, the hnRNAPA0/MY34UE-AS axis regulates the transcription factor MYB through its RNA recognition motifs (RRM2). Heterogeneous Nuclear Ribonucleoprotein A0 (hnRNPA0) upregulates MYB expression by binding to the elncRNA MY34UE-AS, thereby promoting the proliferation and maturation of human erythroleukemic K562 cells [101] (Table 2).

**Table 2 cells-14-01971-t002:** Long non-coding RNAs (LncRNAs) and erythropoiesis. Preponderant lncRNAs involved in erythropoiesis are listed in this table.

lncRNA	Species	Functions	Mechanisms/Regulation	Pathology	References
HIKER	human	increase of erythropoiesis	CSNK2B	Hypoxia, Monge’s disease	[13]
PCED1B-AS1	human	Erythropoiesis differentiation, chromatin accessibility, enucleation	GATA1	ND	[88]
DANCR	human	Chromatin accessibility	RUNX1	ND	[14]
BGLT3	human	Switch γ to β globin	LRF	β-thalassemia and Sickle cell disease	[89]
alncRNA-EC7	mouse	Erythropoiesis differentiation, enucleation	HNRNPU	ND	[10,81,90,91]
Saf	human	Erythropoiesis differentiation, apoptosis	GATA1, KLF1, FAS, NF-κB	ND	[77,93]
EPS	mouse	Apoptosis	ASC/Pycard	Anemia	[85,94]
UCA1	human	Heme regulation	ALAS2, PTBP1	Anemia and heme insufficiency	[95]
Shlnc-EC6	mouse	Enucleation	Rac1 and PIP5K	ND	[86]
NEAT1_1 and NEAT1_2	Human and mouse	Erythroid differentiation	GATA1, KLF2, Myb	Blood cancers and stress erythropoiesis in a mouse model	[96]
elncRNA MY4UE-AS	human	Proliferation and maturation of K562 cells	MYB	ND	[101]
ncRNA-a3	human	Differentiation and maturation of erythroblast, hemoglobin regulation and heme biosynthesis	TAL1	ND	[98]
CpoxeRNA	mouse	Regulation of terminal erythropoiesis and enucleation	CTCF/cohesin complex	ND	[100]

The characteristics of lncRNAs can vary from one species to another, and their dysregulation can impact several molecular actors. This change in gene regulation can then lead to the onset of diseases. Conversely, some diseases can also be caused by the dysregulation of lncRNA expression. HIKER: Hypoxia Induced Kinase-mediated Erythropoietic Regulator; PCED1B-AS1: PC-esterase domain-containing 1B antisense RNA 1; DANCR: Anti-Differentiated Non-Coding RNA; alncRNA-EC7: Bloodlinc; Saf: Fas antisense 1; EPS: lincRNA erythroid pro-survival; UCA1: urothelial cancer associated 1; NEAT1: nuclear paraspeckle assembly transcript 1; GATA1: GATA Binding Protein 1; RUNX1: Runt-related transcription factor 1; LRF: lymphoma/leukemia-related factor; HNRNPU: Heterogeneous nuclear ribonucleoprotein U; KLF1: Kruppel-like factor 1; NF-κB: nuclear factor-kappa B; ALAS2: 5′-aminolevulinate synthase 2; PTBP1: polypyrimidine tract-binding protein 1; Rac1: Ras-related C3 botulinum toxin substrate 1; TAL1: T-cell acute lymphocytic leukemia protein 1; ND: Not determined.

## 3. Conclusions

In recent years, ncRNAs have emerged as pivotal regulators of erythroid development, differentiation, and function. They influence nearly every stage of erythropoiesis, from hematopoietic stem cell commitment to terminal erythroid maturation, by modulating transcriptional programs, chromatin structure, gene expression, apoptosis, and enucleation.

Key genes and pathways involved in erythroid lineage specification, hemoglobin switching, iron metabolism, and response to environmental stressors are regulated by miRNAs. Although less extensively studied, lncRNAs are gaining attention for their roles in epigenetic regulation, alternative splicing, and the fine-tuning of erythropoietic gene networks. They are involved in apoptosis, heme regulation, enucleation, erythroid differentiation, alternative splicing and chromatin regulation.

The diversity and specificity of ncRNA functions in erythropoiesis not only deepen our understanding of RBC biology but also offer promising ways for the development of novel diagnostic biomarkers and therapeutic strategies for hematologic diseases, including anemias and hemoglobinopathies. Future research focusing on the functional characterization and clinical translation of these ncRNAs holds great potential for advancing personalized medicine in hematology.

Very few clinical trials targeting non-coding RNAs in hematopoiesis have been conducted to date, and miRNA-based therapies are still at a very early stage. Currently, experimental interventions involving lncRNAs have been carried out only in pre-clinical models, and these studies were addressed in the manuscript whenever appropriate. Regarding their potential relevance in patients, lncRNAs are strongly linked to the progression of erythropoiesis-related damage, suggesting that therapies aimed at these molecules could help improve this condition. In a murine model, it has been shown that systemically delivered siRNAs targeting the EglN prolyl hydroxylases suppress hepcidin levels specifically in the liver, leading to improved RBC production in models of anemia [102]. In a human clinical trial, the miR-155 inhibitor, MRG-106 (Cobomarsen), was evaluated in a phase 1 clinical trial for various hematologic cancers, including certain leukemias and lymphomas. This led to a phase 2 trial, which was later terminated for business-related reasons [103,104]. Developing RNA-interference–based treatments remains a major challenge, and miRNAs are known to play key regulatory roles in various kidney disorders. Several carriers—such as exosomes, microvesicles, and high-density lipoproteins—can transport functional miRNAs along with lipids, proteins, and mRNAs. These naturally occurring nanovesicles may be harnessed to deliver functional lncRNAs to RBCs. Artificial nanotechnology platforms can also be employed, including 13 nm gold nanoparticles coated with alkylthiol-modified RNA monolayers. The CRISPR/Cas9 system provides another valuable tool in non-coding RNA research, as it can be programmed to target specific genomic sites and permanently disrupt RNA expression. Consequently, lncRNAs, miRNAs, or antisense sequences could be delivered to RBCs depending on therapeutic needs.

## Figures and Tables

**Figure 1 cells-14-01971-f001:**
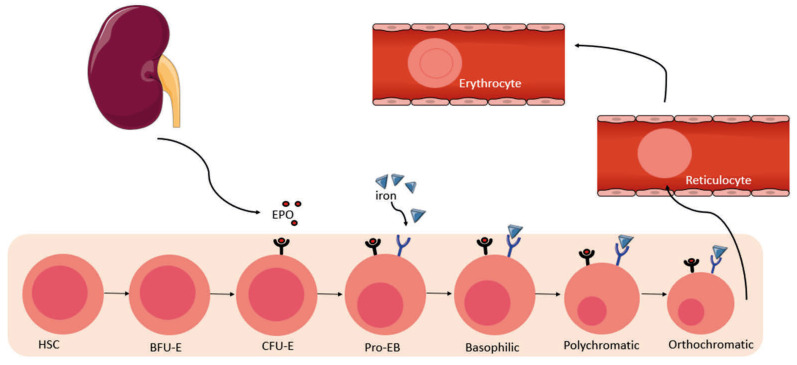
Physiology of erythropoiesis. The kidney releases erythropoietin (EPO), which binds to its receptor on CFU-E and pro-erythroblast cells in the bone marrow, promoting their survival and proliferation, and enabling their progression into the basophilic, polychromatic and orthochromatic stages. Iron also participates in this differentiation by binding to its receptor. Erythroblasts lose their nucleus and then enter the circulation to become reticulocytes and then erythrocytes. HSC—Hematopoietic Stem cells, BFU-E—Erythroid Bursting-Forming Units, CFU-E—Erythroid Colony-Forming Units, Pro-EB—Proerythroblast.

**Figure 2 cells-14-01971-f002:**
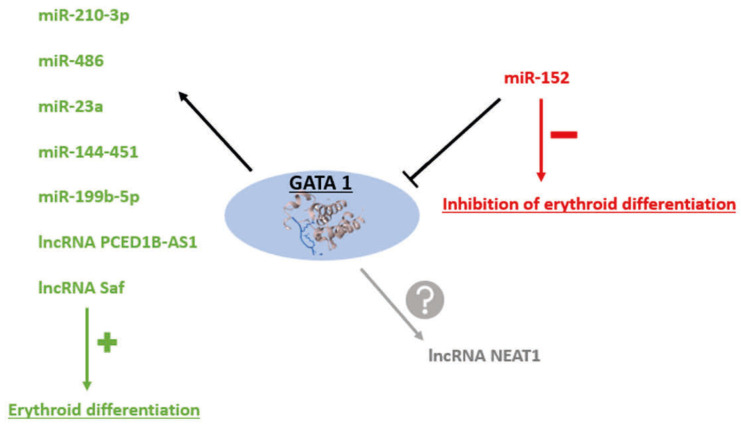
Interactions between non-coding RNA (ncRNA) and the transcription factor Gata Binding Protein 1 (GATA1). GATA1 acts as a transcriptional regulator of several microRNAs (miRNAs) and long non-coding RNAs (lncRNAs), activating their expression to promote erythroid differentiation. However, the expression of GATA1 itself can be negatively regulated, for example, by miR-152. In addition, the lncRNA Nuclear paraspeckle assembly transcript 1 (Neat1) also interacts with GATA1, although the precise nature of this regulation remains to be determined.

**Figure 3 cells-14-01971-f003:**
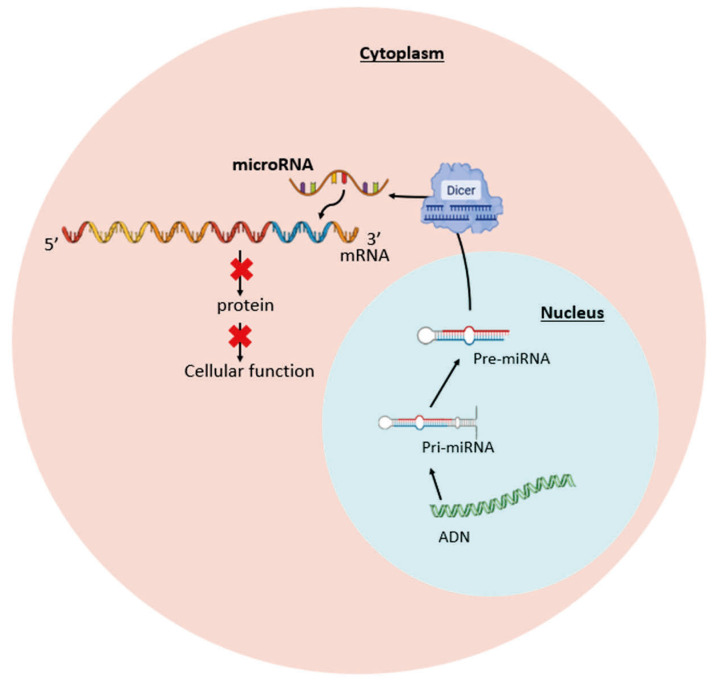
Mechanism of action of microRNAs (miRNAs). In the cytoplasm, miRNAs bind to the 3′-UTR region of the target mRNA. This interaction inhibits protein translation and alters cell functions.

**Figure 4 cells-14-01971-f004:**
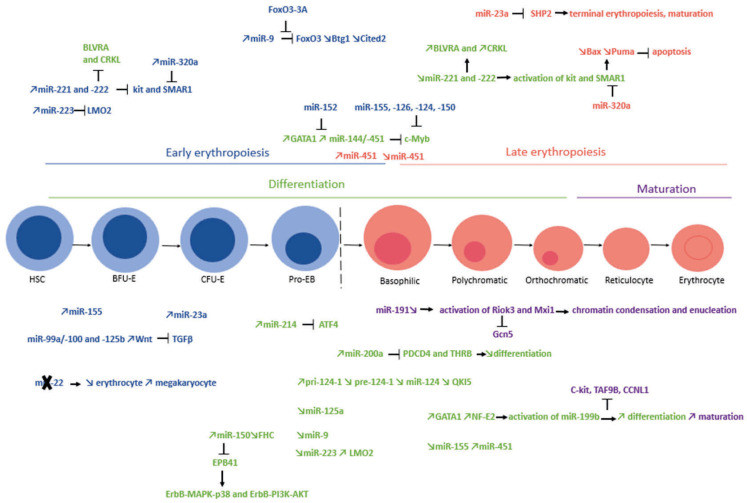
miRNAs and different stages of normal erythropoiesis. The different stages of erythropoiesis are classified into two categories: early erythropoiesis (in blue) and late erythropoiesis (in red). Within these two categories, we refer to erythroid differentiation (from the HSC stage to the orthochromatic stage, in green) and maturation (from the erythroblast stage to the erythrocyte stage, in purple). In each category, miRNAs have specific expression patterns. Some are more involved in early erythropoiesis and differentiation, while others are more involved in late erythropoiesis and maturation. This figure shows the variations in the expression of these miRNAs and the specific targets that they inhibit. The ✕ symbol illustrates that the expression of miR-22 leads to a decrease in the number of erythrocytes and an increase in the number of megakaryocytes. During physiological erythropoiesis, this microRNA is therefore absent.

**Figure 5 cells-14-01971-f005:**
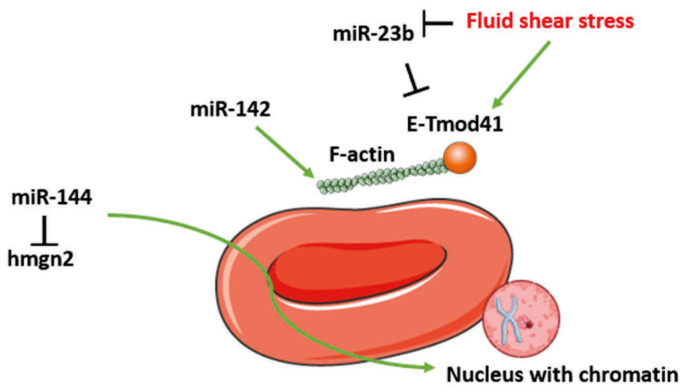
miRNAs and erythrocyte morphology. The morphology of red blood cells (RBCs) is regulated by microRNAs (miRNAs), including miR-23b, miR-142 and miR-144. F-actin dynamics are modulated by miR-142, while miR-23b inhibits the erythrocyte tropomodulin of 41kDa (E-Tmod41). However, miR-23b itself is downregulated by fluid shear stress, leading to the activation of E-Tmod41 and actin filaments (F-actin), which ensures proper organization of the membrane cytoskeleton. Additionally, miR-144 promotes chromatin condensation by targeting the chromatin-associated high mobility group protein N2 (HMGN2), facilitating nuclear extrusion and erythrocyte maturation.

**Figure 6 cells-14-01971-f006:**
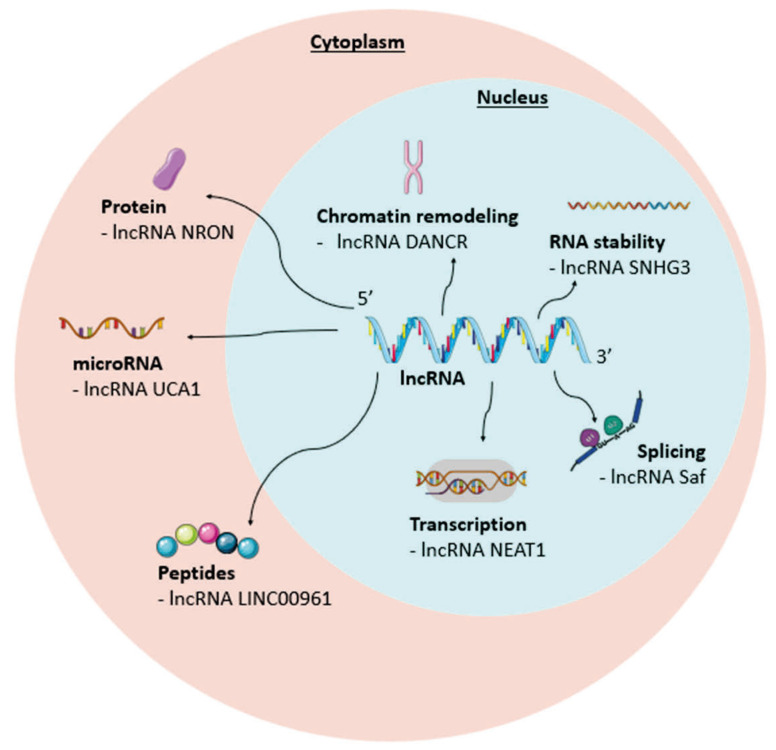
Mechanisms of action of long non-coding RNAs (lncRNAs). Compared to microRNAs (miRNAs), lncRNAs are more diverse and can act in both the cytoplasm and the nucleus of many cells. In the nucleus, they can influence transcription, splicing, RNA stability, and chromatin remodeling. In the cytoplasm, they can code for small peptides and influence the regulation of miRNA expression and certain proteins. DANCR: Anti-Differentiated Non-Coding RNA; Saf: Fas antisense 1; Neat1: nuclear paraspeckle assembly transcript 1; UCA1: urothelial cancer-associated 1; NRON: non-coding repressor of NFAT.

## Data Availability

No new data were created or analyzed in this study.

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
