# Peer review of "The Non-Coding RNome Landscape in Erythropoiesis: Pathophysiological Implications"

_cells, 2025, doi:10.3390/cells14241971_

Round 1
Reviewer 1 Report
Comments and Suggestions for Authors
The Review article is well written and have covered comprehensively the mechanisms for mirs and lncRNAs. I have a few suggestions that might help to make better flow for the article and some cover some important aspects:
- It might be good to write a short paragraph on the interaction of mir and lncRNAs (this may be tie in both better).
- Also in a limitation for lncRNAs-- we should mention about something about uniqueness of lncRNAs to human species also bring a challenge in doing invivo studies due to lack of conservation.
- Some expansion of the translational studies -- are there any studies clinical trials that are successful in this space.
- There are minor typos- "Medicine" and arrow in the first columns of tables etc.
Overall, a good review!
Author Response
Reviewer 1
The Review article is well written and have covered comprehensively the mechanisms for mirs and lncRNAs.
We thank deeply the reviewer for his kind judgement of our paper. His remarks helped us to improve the quality of our manuscript.
I have a few suggestions that might help to make better flow for the article and some cover some important aspects:
- It might be good to write a short paragraph on the interaction of mir and lncRNAs (this may be tie in both better).
We thank the reviewer for this insightful remark. We thus added one new paragraph in sub-section 2.3. LncRNAs and erythropoiesis, page 13, lines 474-493.
« The diverse regulatory interactions of ncRNAs is increasingly recognized. While these interactions remain poorly characterized in hematopoiesis, evidence from other fields demonstrates that miRNAs and lncRNAs can mutually regulate each other through multiple mechanisms. Many lncRNAs act as competitive endogenous RNAs, binding miRNAs to prevent them from repressing target mRNAs. This sponging mechanism allows lncRNAs to indirectly increase the expression of specific protein-coding genes. The miRNAs can also bind to lncRNAs and promote their degradation, reducing the lncRNA’s regulatory activity ( Panni S, Lovering RC, Porras P, Orchard S. Non-coding RNA regulatory networks. Biochim Biophys Acta BBA - Gene Regul Mech. juin 2020;1863(6):194417. 1). Through mutual regulation, miRNAs and lncRNAs form feedback loops that fine-tune cellular signaling pathways. Deregulation of miRNA–lncRNA networks is linked to diseases such as cancer, cardiovascular disorders, and neurological conditions. These networks act as molecular buffers, maintaining gene-expression stability under stress. » ( Ma B, Wang S, Wu W, Shan P, Chen Y, Meng J, et al. Mechanisms of circRNA/lncRNA-miRNA interactions and applications in disease and drug research. Biomed Pharmacother. juin 2023;162:114672.).
Also in a limitation for lncRNAs-- we should mention about something about uniqueness of lncRNAs to human species also bring a challenge in doing invivo studies due to lack of conservation.
We agree with the reviewer and thank him for this clever remark that will improve the paper. As a consequence, we added page 13, lines 506-508, the following sentence.
« It should be mentioned that a limitation for the study of lncRNAs is their uniqueness to human species which brings a challenge in transposing murine findings to human, due to their lack of conservation. ».
- Some expansion of the translational studies -- are there any studies clinical trials that are successful in this space
We deeply agree with the reviewer. We also added other relevant informations, to answer queries from another reviewer. We have combined the two queries in a chapter, Page 19, lines 726-end, that will definitely help to conclude efficiently the manuscript.
Very few clinical trials targeting non-coding RNAs in hematopoiesis have been conducted to date, and miRNA-based therapies are still at a very early stage. Currently, experimental interventions involving lncRNAs have been carried out only in pre-clinical models, and these studies were addressed in the manuscript whenever appropriate. Regarding their potential relevance in patients, lncRNAs are strongly linked to the progression of erythropoiesis-related damage, suggesting that therapies aimed at these molecules could help improve this condition. In a murine model, it has been shown that systemically delivered siRNAs targeting the EglN prolyl hydroxylases suppress hepcidin levels specifically in the liver, leading to improved RBC production in models of anemia (Querbes W, Bogorad RL, Moslehi J, Wong J, Chan AY, Bulgakova E, et al. Treatment of erythropoietin deficiency in mice with systemically administered siRNA. Blood. 30 août 2012;120(9):1916‑22.). In a human clinical trial, the miR-155 inhibitor, MRG-106 (Cobomarsen), which was evaluated in a phase 1 clinical trial for various hematologic cancers, including certain leukemias and lymphomas. This lead to a phase 2 trial, which was later terminated for business-related reasons. ( Kim T, Croce CM. MicroRNA: trends in clinical trials of cancer diagnosis and therapy strategies. Exp Mol Med. 10 juill 2023;55(7):1314‑21.) ( Pal JK, Sur S, Mittal SPK, Dey S, Mahale MP, Mukherjee A. Clinical implications of miRNAs in erythropoiesis, anemia, and other hematological disorders. Mol Biol Rep. déc 2024;51(1):1064.). Developing RNA-interference–based treatments remains a major challenge, and miRNAs are known to play key regulatory roles in various kidney disorders. Several carriers—such as exosomes, microvesicles, and high-density lipoproteins—can transport functional miRNAs along with lipids, proteins, and mRNAs. These naturally occurring nanovesicles may be harnessed to deliver functional lncRNAs to RBCs. Artificial nanotechnology platforms can also be employed, including 13-nm gold nanoparticles coated with alkylthiol-modified RNA monolayers. The CRISPR/Cas9 system provides another valuable tool in non-coding RNA research, as it can be programmed to target specific genomic sites and permanently disrupt RNA expression. Consequently, lncRNAs, miRNAs, or antisense sequences could be delivered to RBCs depending on therapeutic needs.
This growing catalog of ncRNA activities is reshaping the framework of red cell biology and clarifying mechanisms underlying diverse hematologic disorders. As specific ncRNAs continue to be linked to pathological phenotypes—including congenital anemias, ineffective erythropoiesis, and hemoglobinopathies—they are emerging as attractive candidates for precision diagnostics and RNA-based therapeutics. Continued efforts to functionally dissect these molecules and translate their biology into clinical tools will significantly advance the prospects of personalized medicine in hematology. »
- There are minor typos- "Medicine" and arrow in the first columns of tables etc.
We thank deeply the reviewer for having spotted these mistakes. We have cotrected them and read again very carefully the review to avoid other spelling mistakes.
Overall, a good review!
Again, we thank deeply the reviewer for his kind judgement of our paper, and point out again that his clever remarks helped us to improve the quality of our manuscript.
Reviewer 2 Report
Comments and Suggestions for Authors
Major issues
Previous reviews on the topic of miRNAs and lncRNAs in erythroid cell differentiation and maturation should be mentioned and cited in the introduction.
Figure 1 and 4 should include the basophilic, polychromatic, orthochromatic erythroblast stages, as these are very well established since a long time, and also with respect to regulation by the RNome.
In Figure 2 it is not clear what the arrow pointing to lncRNA NEAT1 means. Interaction, regulation, etc.?
Although not further discussed as such, Figure 3 is quite simplistic; additional details could be provided in the figure.
In table 1 and 2 (also in the text), it should be made clear (using different column or color) whether the data correspond to actual functional and mechanistic insights (based for example on loss-of-function studies) or whether the results just reflect correlation (for example up- or down regulation at specific stage of erythropoiesis, etc.).
The miRNA-191 should be included in the review/tables/figures. Please refer to its role in enucleation (Zhang et al., Genes Dev 2011; doi: 10.1101/gad.1998711)
Lines 441- Many more lncRNAs have been identified. Please cite Kaur et al. 2024 doi: 10.1101/2024.10.29.620654 (for example Figure 1E in this article). Consequently, please also correct wrong statement on lines 605-606.
There should be a separate section on therapeutic applications (just before the conclusion). Please discuss possible translational applications, ongoing clinical trials, etc. with respect to the RNome, especially considering recent success with RNA therapeutics (vaccines, antisense oligonucleotides-based drugs, etc.).
The conclusion section should be expanded and be less generic.
Minor issues
Lines 59-65. Mention the importance of the cytoskeleton to maintain a biconcave shape.
Lines 82-84: Is the expression of the ncRNA cause or just an effect?
Please clarify the following statement: “A number of miRNAs may have their expression altered in the presence of IS…” “May” is rather vague, do they or do they not?
Statement lines 247-249: is there proof/reference(s) for this statement or is it just speculation?
Lines 274-275: avoid “may”; is it or is it not?
Lines 320- This is known as the cross antagonistic model of regulation of hematopoiesis by competing transcriptional regulators. Please mention and cite references.
Do any of the miRNAs carry out their function in erythropoiesis through the peptides they encode?
Comments on the Quality of English LanguageIn Figure 1 and 4 french words are used (fer, erythroblaste), which should be translated into English.
There are a few spelling mistakes (coloborrative, futhermore, etc.). The entire text should be verified.
The term “et al.” is written in various forms. It should be in italics and with a period: “et al.”
Sentences should not start with small letters, for example miRNA, lncRNA. Hence “miRNAs play an essential role in regulating all stages of hematopoiesis.” could be changed to “It is well established that miRNAs play…”
“…this alternative system is only temporary.” should be “…this alternative system is active only temporary.”
“UT are very numerous…” should be “UTs are very numerous...”
“have been better studied…” should be “have been studied in more detail…”
“most ncRNA do not code proteins.” should be “, most ncRNA do not code for proteins.”
Lines 108-109: avoid double brackets, or use different types of brackets: (, [.
Line 558 should be: Like other cellular processes, apoptosis (or programmed cell death) is tightly regulated.
Author Response
Reviewer 2
We thank deeply the reviewer for his kind judgement of our paper. His remarks helped us to deeply improve the quality of our manuscript.
Major issues
Previous reviews on the topic of miRNAs and lncRNAs in erythroid cell differentiation and maturation should be mentioned and cited in the introduction.
We thank the reviewer for this remark, and agree with him. Several reviews have indeed already addressed the role of miRNAs and lncRNAs in the differentiation and maturation of erythroid cells, thereby summarizing the current knowledge on the function of these ncRNAs. Our review aims to complement and update this litterature by integrating recent discoveries and simultaneously addressing the roles of both miRNAs and lncRNAs in erythropoiesis.
We have added in the introduction the following sentences, page 4, lines 114
« Several reviews have already addressed the role of miRNAs and/or lncRNAs in the differentiation and maturation of erythroid cells. (Lawrie CH. microRNA expression in erythropoiesis and erythroid disorders. Br J Haematol. juill 2010;150(2):144‑51 ; Azzouzi I, Schmugge M, Speer O. MicroRNAs as components of regulatory networks controlling erythropoiesis. Eur J Haematol. juill 2012;89(1):1‑9 ; Wang FY, Gu ZY, Gao CJ. Emerging role of long non-coding RNAs in normal and malignant hematopoiesis. Chin Med J (Engl). 20 févr 2020;133(4):462‑73 ; Li Y, Zhang H, Hu B, Wang P, Wang W, Liu J. Post-transcriptional regulation of erythropoiesis. Blood Sci. juill 2023;5(3):150‑9 ; Kulczyńska K, Siatecka M. A regulatory function of long non-coding RNAs in red blood cell development. Acta Biochim Pol [Internet]. 4 mars 2017 [cité 2 déc 2025];63(4) ; Xu C, Shi L. Long non-coding RNAs during normal erythropoiesis. Blood Sci. oct 2019;1(2):137‑40.). Our review aims to complement and update this literature by integrating recent discoveries and simultaneously addressing the roles of both miRNAs and lncRNAs in erythropoiesis. »
«
Figure 1 and 4 should include the basophilic, polychromatic, orthochromatic erythroblast stages, as these are very well established since a long time, and also with respect to regulation by the RNome.
We thank the reviewer for the comment. The basophilic, polychromatic, and orthochromatic stages have been added to Figures 1 and 4. The figure legends have been changed accordingly. However, to the best of our knowledge and after several checks, we have not identified any microRNAs specifically associated with these stages. Although these stages and their regulation are increasingly well understood, it remains difficult to attribute a specific microRNA to a particular stage.
In Figure 2 it is not clear what the arrow pointing to lncRNA NEAT1 means. Interaction, regulation, etc.?
The lncRNA NEAT1 has been shown in grey with a simple arrow because its regulation by GATA1 is not yet well defined. A question mark has therefore been added to the figure. This justification is stated in the figure legend.
Although not further discussed as such, Figure 3 is quite simplistic; additional details could be provided in the figure.
We thank the reviewer for their comment. We have therefore added several elements to our figure, notably some aspects of miRNA biogenesis.
In table 1 and 2 (also in the text), it should be made clear (using different column or color) whether the data correspond to actual functional and mechanistic insights (based for example on loss-of-function studies) or whether the results just reflect correlation (for example up- or down regulation at specific stage of erythropoiesis, etc.).
We agree with the reviewer and added as a consequence an additional sentence that will help to distinguish the two types of study, line 238 in Table Legend. « All studies were mechanistic. »
In addtiion, we added the following sentence line 406. « Phannasil et al., based on a correlation study, highlighted that the expression level of this miRNA varies »
The miRNA-191 should be included in the review/tables/figures. Please refer to its role in enucleation (Zhang et al., Genes Dev 2011; doi: 10.1101/gad.1998711)
We thank the reviewer for suggesting this article, which provides a valuable addition to our literature review. miR-191 has been incorporated into Figure 4 and mentioned in the main text, page 8, lines 277-281 « The decrease in miR-191 is essential for chromatin condensation and enucleation. In the murine model of erythropoiesis, miR-191 expression decreases from the CFU-E stage onward. This reduction allows the activation of its targets Riok3 and Mxi1, leading to the inhibition of the histone acetyltransferase Gcn5, thereby facilitating chromatin condensation and enucleation. » ( Zhang L, Flygare J, Wong P, Lim B, Lodish HF. miR-191 regulates mouse erythroblast enucleation by down-regulating Riok3 and Mxi1. Genes Dev. 15 janv 2011;25(2):119‑24. )
Lines 441- Many more lncRNAs have been identified. Please cite Kaur et al. 2024 doi: 10.1101/2024.10.29.620654 (for example Figure 1E in this article). Consequently, please also correct wrong statement on lines 605-606.
We thank the reviewer for pointing out this really recent and interesting publication. We have modified the text accordingly, and cited the corresponding reference.
Line 455 : Thanks to improved database analyses, a study published in 2024 identified 140 268 new human lncRNA transcripts in the GENCODE database. The new number has been added in the manuscript.
Lines 640: The sentence : « Although they are fewer than miRNAs, and less well known, their role is equally important » has been deleted and the following has been added to the text : « Indeed, ongoing technological progress is already enabling the discovery of novel lncRNAs, suggesting that some of them may also play a role in erythropoiesis. ». The Kaur et al. Paper has been added to the text and bibliography.
There should be a separate section on therapeutic applications (just before the conclusion). Please discuss possible translational applications, ongoing clinical trials, etc. with respect to the RNome, especially considering recent success with RNA therapeutics (vaccines, antisense oligonucleotides-based drugs, etc.). The conclusion section should be expanded and be less generic.
Answer. We gratefully agree with the reviewer. As a result, we added the following paragraph page, in the perspectives section. We have answered in a chapter, Page 19, lines 726-end, that will definitely help to conclude efficiently the manuscript, and we hope thus that the conclusion section is less generic.
Very few clinical trials targeting non-coding RNAs in hematopoiesis have been conducted to date, and miRNA-based therapies are still at a very early stage. Currently, experimental interventions involving lncRNAs have been carried out only in pre-clinical models, and these studies were addressed in the manuscript whenever appropriate. Regarding their potential relevance in patients, lncRNAs are strongly linked to the progression of erythropoiesis-related damage, suggesting that therapies aimed at these molecules could help improve this condition. In a murine model, it has been shown that systemically delivered siRNAs targeting the EglN prolyl hydroxylases suppress hepcidin levels specifically in the liver, leading to improved RBC production in models of anemia (Querbes W, Bogorad RL, Moslehi J, Wong J, Chan AY, Bulgakova E, et al. Treatment of erythropoietin deficiency in mice with systemically administered siRNA. Blood. 30 août 2012;120(9):1916‑22.). In a human clinical trial, the miR-155 inhibitor, MRG-106 (Cobomarsen), which was evaluated in a phase 1 clinical trial for various hematologic cancers, including certain leukemias and lymphomas. This lead to a phase 2 trial, which was later terminated for business-related reasons. ( Kim T, Croce CM. MicroRNA: trends in clinical trials of cancer diagnosis and therapy strategies. Exp Mol Med. 10 juill 2023;55(7):1314‑21.) ( Pal JK, Sur S, Mittal SPK, Dey S, Mahale MP, Mukherjee A. Clinical implications of miRNAs in erythropoiesis, anemia, and other hematological disorders. Mol Biol Rep. déc 2024;51(1):1064.). Developing RNA-interference–based treatments remains a major challenge, and miRNAs are known to play key regulatory roles in various kidney disorders. Several carriers—such as exosomes, microvesicles, and high-density lipoproteins—can transport functional miRNAs along with lipids, proteins, and mRNAs. These naturally occurring nanovesicles may be harnessed to deliver functional lncRNAs to RBCs. Artificial nanotechnology platforms can also be employed, including 13-nm gold nanoparticles coated with alkylthiol-modified RNA monolayers. The CRISPR/Cas9 system provides another valuable tool in non-coding RNA research, as it can be programmed to target specific genomic sites and permanently disrupt RNA expression. Consequently, lncRNAs, miRNAs, or antisense sequences could be delivered to RBCs depending on therapeutic needs.
This growing catalog of ncRNA activities is reshaping the framework of red cell biology and clarifying mechanisms underlying diverse hematologic disorders. As specific ncRNAs continue to be linked to pathological phenotypes—including congenital anemias, ineffective erythropoiesis, and hemoglobinopathies—they are emerging as attractive candidates for precision diagnostics and RNA-based therapeutics. Continued efforts to functionally dissect these molecules and translate their biology into clinical tools will significantly advance the prospects of personalized medicine in hematology.
Minor issues
Lines 59-65. Mention the importance of the cytoskeleton to maintain a biconcave shape.
We thank the reviewer for this insightful remark. As a result, we added the sentence « The cytoskeleton is instrumental to maintain the characteristic biconcave shape of the RBC. » page 11 lines 423-424
Lines 82-84: Is the expression of the ncRNA cause or just an effect?
Thanks for pointing out this interesting question. We consider that the mechanistic studies (see Table 1) show that the expression of the ncRNAs is causal in the orchestration of the proper progression of the process.
As a result, we added in the manuscript the following sentence, page 3 lines 85-87.
« Under physiological conditions, all the different actors are perfectly synchronized and mechanistic studies (see Table 1) show that the expression of the ncRNAs is causal in the orchestration of the proper progression of the process. »
Please clarify the following statement: “A number of miRNAs may have their expression altered in the presence of IS…” “May” is rather vague, do they or do they not?
They do. We have for example shown that concerning miR-126 and miR-223 (ref 38). We thus corrected this in the manuscript.
Statement lines 247-249: is there proof/reference(s) for this statement or is it just speculation?
Lines 274-275: avoid “may”; is it or is it not?
We agree with the reviewer and this was corrected accordingly in the manuscript.
Lines 320- This is known as the cross antagonistic model of regulation of hematopoiesis by competing transcriptional regulators. Please mention and cite references.
We agree with the reviewer and this point was raised by another reviewer. We have added the following paragraph to answer both queries in sub-section 2.3. LncRNAs and erythropoiesis, page 13, lines 482-493.
« The diverse regulatory interactions of ncRNAs is increasingly recognized. While these interactions remain poorly characterized in hematopoiesis, evidence from other fields demonstrates that miRNAs and lncRNAs can mutually regulate each other through multiple mechanisms. Many lncRNAs act as competitive endogenous RNAs, binding miRNAs to prevent them from repressing target mRNAs. This sponging mechanism allows lncRNAs to indirectly increase the expression of specific protein-coding genes. The miRNAs can also bind to lncRNAs and promote their degradation, reducing the lncRNA’s regulatory activity ( Panni S, Lovering RC, Porras P, Orchard S. Non-coding RNA regulatory networks. Biochim Biophys Acta BBA - Gene Regul Mech. juin 2020;1863(6):194417. 1). Through mutual regulation, miRNAs and lncRNAs form feedback loops that fine-tune cellular signaling pathways. Deregulation of miRNA–lncRNA networks is linked to diseases such as cancer, cardiovascular disorders, and neurological conditions. These networks act as molecular buffers, maintaining gene-expression stability under stress. » ( Ma B, Wang S, Wu W, Shan P, Chen Y, Meng J, et al. Mechanisms of circRNA/lncRNA-miRNA interactions and applications in disease and drug research. Biomed Pharmacother. juin 2023;162:114672.).
Do any of the miRNAs carry out their function in erythropoiesis through the peptides they encode?
This is an interesting question. Some lncRNAs have indeed a coding sequence that will enable producion of small peptides. This has been described in several fields, however , to the best of our knowledge there is no relevant literature in the erythropoiesis field.
In Figure 1 and 4 french words are used (fer, erythroblaste), which should be translated into English.
We thank the reviewer for having spotted these mistakes and thank him greatly. The figures have been corrected.
There are a few spelling mistakes (coloborrative, futhermore, etc.). The entire text should be verified.
The two mistakes mentionned have been corrected. Thanks for spotting them.
The term “et al.” is written in various forms. It should be in italics and with a period: “et al.”
All the « et al. » have been corrected in the text. Thanks for spotting them.
Concerning the « et al. » in the reference list, we will let the journal do the adjustements according to its policy.
Sentences should not start with small letters, for example miRNA, lncRNA. Hence “miRNAs play an essential role in regulating all stages of hematopoiesis.” could be changed to “It is well established that miRNAs play…”
This has been corrected throughout the manuscript. Thank you for spotting that.
“…this alternative system is only temporary.” should be “…this alternative system is active only temporary.”
This has been corrected. Thank you for spotting that.
“UT are very numerous…” should be “UTs are very numerous...”
This has been corrected throughout the manuscript. Thank you for spotting that.
“have been better studied…” should be “have been studied in more detail…”
“most ncRNA do not code proteins.” should be “, most ncRNA do not code for proteins.”
This has been corrected. Thank you for spotting that.
Lines 108-109: avoid double brackets, or use different types of brackets: (, [.
We appreciate the comment, and we will let the journal do the adjustements according to its policy.
Line 558 should be: Like other cellular processes, apoptosis (or programmed cell death) is tightly regulated.7
This has been corrected, thank you for having spotted the mistake.